# Internationalization of Higher Education in China with Spain: Challenges and Complexities

Yinglong Qiu [1,*], Adela García-Aracil [1] and Rosa Isusi-Fagoaga [2]

1  Institute of Innovation and Knowledge Management, INGENIO (CSIC-UPV), Universitat Politècnica de València, 46022 Valencia, Spain; agarcia@ingenio.upv.es
2  University Institute of Educational Creativity and Innovation (IUCIE), Universitat de València, 46022 Valencia, Spain; rosa.isusi@uv.es
*  Correspondence: yqiu2@upv.edu.es

**Abstract:** The coronavirus 2019 pandemic has influenced the internationalization of higher education, reflecting its broader impacts on the economic, geopolitical, and technological development of countries. Governments have prioritized the internationalization of higher education post-pandemic to generate income and create a sustainable economy by cultivating foreign language talents. Although there are studies analyzing the internationalization of higher education through the students' mobility, in our case, there is a scarcity of studies on the internationalization between China and Spain; in particular, Spanish universities lack references to attract more Chinese students. To bridge this gap in the literature, we conducted semi-structured interviews with eight experts in the implementation of internationalization in higher education from four universities in China and Spain. The findings reveal that internationalization is associated with promoting Chinese regional development, and their students can access better quality education. Key motivators for Chinese students in Spain include the improved world ranking of universities and the demand for more courses in English. Although the biggest obstacle to cooperation between both countries is communication, a trusted intermediary could overcome part of this problem. Moreover, the use of new technologies could facilitate sustainable internationalization and regional development.

**Keywords:** higher education; internationalization; socio-economic development; China; Spain

## 1. Introduction

Higher education is one of many economic drivers, playing a crucial role by contributing to national and institutional income. Educational exports, or transnational education, impart education to foreign students through franchising agreements and by establishing affordable domestic campuses to facilitate the international mobility of incoming students [1]. Educational exports contribute significantly to the economy and help build international relationships [2]. However, economic problems have considerably reduced the public resources available to many higher education institutions [3], with many universities regarding internationalization as a viable alternative or complement to domestic funding [4]. The growing demand for transnational higher education has resulted in significant education exporters formulating various development strategies and specific measures to strengthen their exports and expand their market share in transnational higher education [5].

Along with globalization, the internationalization of higher education (hereafter INHE) promotes the development of students and the local economy [5,6]. Several countries have updated their strategies to promote the INHE; for example, in the United Kingdom, the Department for Education published an international education strategy for all sectors at all levels of education in 2019, updating it on 6 February 2021, to not only support the recovery of international education during and after the coronavirus 2019 (COVID-19) pandemic but

also boost economic growth [2]. Another example of internationalization is the Australian Strategy for International Education 2021–2030, which outlines a sustainable growth path for Australia's international education sector, focusing on diversification, meeting national skills needs, placing students at the center, and bolstering global competitiveness [7,8]. In March 2023, Spain, as one of the countries of the European Union, approved the Organic Law of the University System, establishing, for the first time, a title dedicated to internationalization with the purpose of not only promoting a quality university system with agile and reliable evaluation mechanisms but also planning and developing internationalization strategies by different public administrations and universities, together with the creation of interuniversity alliances and participation in international, supranational, and Euro regional projects. When the mobility of the university community is promoted, incentives are provided for doctorates in international co-supervision, and public administrations are urged to eliminate obstacles to attracting international talent, speeding up and facilitating procedures for the recognition and homologation of degrees, admission to universities, and migration [9]. Without exception, China, as an emerging economy, attaches great importance to the implementation of the strategy of internationalization of education to meet the challenges of the globalization era [10], and it published the "Outline of the Medium- and Long-Term Scientific and Technological Development Plan of Higher Education Institutions (2021–2035)" and the "Fourteenth Five-Year Plan for Scientific and Technological Development of Higher Education Institutions" in 2021 to accelerate and expand international cooperation in the new era, optimizing global readiness and strengthening local development in talent cultivation and scientific research [11]. Given these ambitious national goals, the economic rationale for establishing, legitimizing, and maintaining international relations is crucial, as the INHE promotes local development [12].

The INHE is also known as "cross-border education" or "transnational education". It refers to students traveling abroad to undertake a course of study or enjoy educational services (including distance learning) at different types of higher education institutions, ultimately having the opportunity to obtain a degree from that country [13]. The literature reveals that the INHE has a positive effect on students' professional and personal development in terms of enhanced recognition for holding a global university degree, improved foreign language skills, communicative experience in an intercultural environment, communication internships in international organizations, and employment opportunities in international companies, among others; it also positively affects the home country when students return with valuable knowledge and experiences [14–16]. In this context, the INHE may serve to achieve the United Nations' Sustainable Development Goal (SDG) target for education [17] by 2030: Ensure inclusive and equitable quality education and promote lifelong learning opportunities for all.

Thus, higher education institutions should conduct further studies to effectively cope with the dynamic global landscape and investigate university staff's perceptions and experiences [18,19]. Analyzing the evolution of this particular research domain is important because it allows experts to explore trends based on temporal issues, geographical regions, institutional support, primary sources, subfields, and research topics [20]. Thus, in the context of the 50th anniversary of the establishment of diplomatic relations between China and Spain [21], we conduct an in-depth investigation into the INHE in both countries to find the most suitable cross-border higher education sustainability models after the COVID-19 era such as Chinese-medium programs, international campuses, non-official bachelor's degrees, or student exchange programs.

*1.1. The INHE in the Age of Digitalization: Positive or Negative?*

The INHE contributes to the economic sustainability of participating universities [5,6]. Still, academic mobility is widely recognized as causing energy depletion and an unsustainable and environmentally unfriendly phenomenon not conducive to global sustainable development [22,23]. A study by the University of the Netherlands of students and faculty in international academic exchanges through international travel revealed that transport

accounted for 40 to 90 percent of carbon emissions [24]. Therefore, various innovative strategies to achieve sustainable internationalization have emerged due to the continuous development of education and technology [25]. Following the COVID-19 pandemic, the mode of the INHE has gradually evolved, leveraging new technologies through virtual reality, augmented reality, and the utilization of the metaverse; using digital platforms that favor virtual exchange to promote international educational experiences [4,25]; and embracing the pedagogical method—the model of virtual exchange [26,27]—a particularly effective approach to achieving sustainable internationalization [28]. Driven by external factors such as the new generation of smart technologies, the digital transformation of INHE has led to development opportunities and challenges, especially in the context of the COVID-19 pandemic [29]. The development trend in higher education reflects digital transformation along with technological innovations such as 5G, artificial intelligence, big data, the Internet of Things, and the metaverse, all of which have transcended geography and borders [30]. For example, due to the use of digitalization and information and communication technology (ICT) tools for communication, more and more students from different countries can come together and learn a new language or skill through virtual communication [31]. However, many students rely heavily on online translation tools, whose efficacy is limited in university academia due to inadequate vocabulary development in academic disciplines [32]. Thus, there is a need to better understand the multifaceted influence of ICT on the INHE [33].

However, sustainable internationalization is not limited to environmental issues but also relates to inclusive and equitable quality education (SDG4), gender equality (SDG5), and decent work and economic growth (SDG8) [17,34]. Many experts believe that digital transformation in education has proven its viability during COVID-19 [35,36]. These benefits extend beyond physical study in classrooms and include teamwork, freedom from constraints by immigration policies, joint programs between universities, opinions of experts from around the world, reduced expenditure, virtual mobility, and the democratization of access and learning processes [37,38]. Regarding learning outcomes, using digital tools helps students succeed by developing their capacity for realistic goal-setting, self-monitoring, self-reflection, self-awareness, and collaboration [39]. In addition, the problem of unequal levels of technological advancement in different geographic regions and economic strata of society underscores the problem of the pre-pandemic digital divide [40]. In other words, the inequalities in access to and consumption of ICTs and digital technologies impact individuals, organizations, and countries [41]. Thus, whether developed or poor, the digitalization of education is influenced by economic development, the ability to purchase equipment, Internet access, infrastructure limitations, and several other issues [42].

As a result, experts have not been optimistic about the INHE through ICT due to challenges such as unequal or inequitable access to educational technologies [43], increased anxiety [44], and data protection issues. Data have become a key element in educational management and policy-making research in all countries [45–47]. The digitalization of the INHE has attracted increasing research and courted controversy. Thus, we must delve deeper into the implications of digitalizing the INHE [43]. Overall, digitalization is an effective model for INHE sustainability despite the inevitable development-related problems that must be solved.

### 1.2. The INHE and the Promotion of Local Development

In the last three decades, the INHE has evolved into international trade driven by historical, economic, and geopolitical factors [48]. With the increasing demand for transnational higher education, major education-exporting countries have developed various development strategies and concrete initiatives to strengthen their higher education exports and increase their market share in the INHE [49]. For example, international students in Canada contributed an estimated $21.6 billion to Canada's GDP in 2018 and supported nearly 170,000 jobs for Canada's middle class [50]. However, with the rapid increase in international economic globalization, the demand for foreign language talents in societies

demonstrates a diversified trend. Thus, the training mode of foreign language talents should be adapted to meet local development needs [51]. With approximately 496 million native Spanish speakers, Spanish is not only the second most spoken language in the world after Mandarin Chinese but also the second language in terms of international communication, according to a report published by Instituto Cervantes [52]. This has fueled a wave of Spanish learning worldwide, making it one of the most studied foreign languages after English and French [53]. The number of colleges and universities offering Spanish programs in China exceeds 140, resulting in more Chinese students seeking to study in Spanish-speaking countries [54]. In Spain, this upward trend is accentuated only if Chinese students who do not reside in Spain are considered: between 2016 and 2020, their number increased by 106%, from 1324 to 2737. They are now among the largest groups of foreigners in Spanish universities. In the academic year 2019–2020, there were 12,571 students of Chinese nationality, behind only Italy, Ecuador, France, and Colombia [55]. However, Scopus and Web of Science searches for the keywords "internationalization AND China AND Spain AND Education" revealed no article on the INHE in China and Spain. Therefore, this study focuses on the INHE in China and Spain.

Despite the dwindling cases of COVID-19, the INHE and student mobility face enormous pressures and challenges [56]. According to the Organization for Economic Cooperation and Development (OECD), the number of international students is expected to reach over 8 million by 2025 [57,58]. The UNESCO Global Education Monitoring Report notes that in 2018, China was the leading country of origin for international students worldwide [59]. UNESCO's Institute of Statistics [60] indicates that, in the realm of education, considering inbound internationally mobile students by country of origin—both sexes included—China remains the top global source of international students before and even during the dwindling COVID-19 pandemic. The top 10 destinations of Chinese higher education students abroad are the United States, Japan, Australia, the United Kingdom and Northern Ireland, Canada, the Republic of Korea, France, Hong Kong (China), New Zealand, and Macao (China) [60]. Relevant research reports of the INHE in the Study Abroad Services industry reveal that, compared with the rapid development from 2014 to 2019, China's Study Abroad Services industry reflected slow development from 2021 to 2023 due to COVID-19 but is expected to grow at a high rate from 2024, reaching a market size of 467.9 billion yuan and nearly 600 billion yuan in 2024 and 2025, respectively [6]. According to the Economic Impact of International Students in Spain, 2020, in the academic year 2018–2019, 1,044,898 international students enrolled in Spanish institutions, with their overall economic impact on the Spanish economy amounting to €3,795,740,732 [61].

Furthermore, due to the scarcity of quality educational resources [62], with the international campus model, while universities play key roles of education and training in local economies around the world, little is known about the significant impact that international campuses have on local training and the local economy [63]. For students, studying at an international campus is highly desirable due to various factors, including the cost of living in their home country, government stability, modern conveniences, cultural and religious proximity, lack of discrimination, and reputation for high quality or international recognition of educational qualifications [64,65]. According to statistics from the US Cross-Border Education Research Team (C-BERT), as of 20 November 2020, there were 306 international campuses worldwide, with the United States, the United Kingdom, France, Russia, and Australia being major exporters of international campuses. China, the United Arab Emirates, Singapore, Malaysia, and Qatar are the largest importers of international campuses [66]. Spain has six imported and two exported international campuses [66]. A similar dual degree model is being developed in China, also known as "Chinese-Foreign Cooperation in Running Schools". China already has 21 projects in collaboration with Spain [67]. This education model helps improve competencies (knowledge, skills, and attributes) while fostering social mobility [68].

*1.3. Regulation and History of China's INHE with Spain*

China's INHE with Spain dates to the 16th century, coinciding with the interest among Chinese and Spanish speakers in each other's languages [69,70]. In 1973, the Kingdom of Spain and the People's Republic of China established full diplomatic relations [71]; 2023 marked 50 years of friendship between the People's Republic of China and the Kingdom of Spain [72].

In 2001, China participated in the World Trade Organization, officially opening its INHE market [73]. On 7 April 1981, an agreement on cultural, educational, and scientific cooperation between Spain and the People's Republic of China was signed in Madrid [49]. On 21 October 2007, an agreement on the Recognition of Degrees and Diplomas between the governments of the Kingdom of Spain and the People's Republic of China was signed in Beijing [74]. On 14 June 2022, the Chinese Service Center for Scholarly Exchange (CSCSE) published new regulations for official bachelor's, master's, and doctoral degrees with a supplementary degree or certificate that must justify the urgent need for recognition. If there is no justification, the degree will only be validated with the final form of the official degree [75]. On 4 November 2022, CSCSE declared that, from 19 October 2023, Spain's own master's degree would be called Master of Lifelong Learning, which would not be validated by CSCSE in China [76]. In 2023, the two countries signed an agreement on the Executive Plan for Cooperation in Education for 2023–2026 [77].

An overview of Spanish internationalization policies reveals that Organic Law 6/2001 on Universities, as amended by Organic Law 4/2007, establishes that the government, autonomous communities, and universities, within the scope of their respective competencies, are responsible for contributing to the European dimension of education, adopting the necessary measures to achieve the integration of the Spanish university system within the European Higher Education Area. Organic Law 3/2022, which aims for the convergence of the vocational training systems of the European Union and the Global South, favoring internationalization and transnational mobility [78], reveals the commitment of the Spanish education system to the European and international dimension of education and assumes the challenge of opening it up to the outside world [79]. Therefore, Spain is politically open to the development of the INHE. Data on foreigners with authorization to stay for studies from 2023 reveal that, in 2022, the number of foreign students coming to Spain reached 63,030, of which 5304 were Chinese students. The largest number of foreign students in Spain—6990—came from the United States [80].

## 2. Objectives

In previous research on the INHE in China and Spain, most researchers have focused on Spanish language teaching [81,82] or the characteristics and status of the higher education system in China and Spain [83,84]. However, few have studied the impact and importance of the INHE from the perspective of technological innovation [25] and current obstacles to international cooperation, factors facilitating the development, and specific needs of the INHE in the post-COVID-19 era.

Therefore, we intend to introduce a conceptual analysis of the relationship between globalization and the INHE in China and Spain, especially to encourage more Chinese students to attend Spanish universities, thus promoting its development and determining future trends. Therefore, this article has three main objectives:

1. To understand the factors influencing Chinese students to study in Spain, the obstacles, and the need for collaboration between Chinese and Spanish universities to promote the INHE between China and Spain.
2. To explore the INHE and its impact on contributing to the local development of China and Spain.
3. To understand experts' opinions on cross-border distance education and future trends in the INHE in China and Spain by leveraging new technologies, driven by the observation that the INHE has resulted in several innovative initiatives due to technology development [25].

## 3. Methodology

This study aims to determine the perception of collaboration models, identify problems, and explore perspectives regarding international cooperation between China and Spain, especially in technological change, and how international cooperation can contribute to local development.

To meet the objective of this study, semi-structured interviews were conducted with experts to allow them to fully express their thoughts and opinions in response to our questions [85,86]. The participants came from international and diverse backgrounds. They were experts in international relations with Chinese and Spanish universities. Moreover, they boast many real success stories on the INHE between China and Spain (see Table 1).

**Table 1.** Participants' details.

| Expert | Sex | Location of University |
|---|---|---|
| Expert ES A | Woman | Catalonia and Madrid Region |
| Expert ES B | Man | Valencia Region |
| Expert ES C | Man | Aragon Region |
| Expert ES D | Man | Castilla de La Mancha Region |
| Expert CN E | Man | Jinan, Shandong Province |
| Expert CN F | Man | Xuzhou, Jiangsu Province |
| Expert CN G | Woman | Shenzhen, Guangdong Province |
| Expert CN H | Woman | Shanghai |

Source: Authors.

However, in the process of searching for the interviewed experts, we encountered two significant difficulties: first, there are fewer experts in international cooperation between Chinese and Spanish universities, as most of them are with English-speaking countries such as the United States, United Kingdom, Canada, or Australia; second, the international relations office is less involved and does not have sufficient knowledge of the particularities of the INHE between China and Spain. Thus, our final selection of respondents followed the recommendation of the international relations office because these experts are responsible for INHE between China and Spain; they have many successful and unsuccessful cases of cooperation between China and Spain.

To obtain valid responses that explain differences or similarities, we conducted a comparative analysis of experts' thoughts and opinions in response to our questions [87]. These experts represent four universities from Spain and China. From each autonomous community and province, we selected one university with notable INHE experience between China and Spain. The selected universities in Spain include one private university located both in the Catalonia and Madrid regions and three public universities in the Valencian Region, Aragon Region, and Castilla-La Mancha Region. China's public universities are in Jinan, Shandong Province; Xuzhou, Jiangsu Province; and Shenzhen, Guangdong Province. China's private university is located in Shanghai City. All of them are top universities worldwide, according to Shanghai Ranking's Academic Ranking of World Universities (ARWU) [88]. Based on data obtained from experts and information published on university websites, we know the number of Spanish universities collaborating with Chinese universities and the number of Chinese universities collaborating with Spanish universities. Also, to illustrate the economic variability of the regions where the universities under study are located, a comparison was made between regions with different economic levels. At the same time, we listed the annual GDP of the regions where the universities are located. These data are shown in Table 2.

**Table 2.** Data of selected universities.

| Spanish Universities | Classification | Autonomous Community/Province | GDP Annual 2022 | Shanghai Ranking ARWU | Number of Collaboration Agreements with Chinese Universities |
|---|---|---|---|---|---|
| Catalonia and Madrid Region | Private | Autonomous Community of Madrid and Catalonia | Madrid: 261.713 M€ Catalonia: 255.154 M€ | 1001+ | 5 |
| Valencian Region | Public | Autonomous Community of Valenciana | 126.416 M€ | 401–500 | 11 |
| Aragon Region | Public | Community of Aragon | 41.763 M€ | 601–700 | 23 |
| Castilla de La Mancha Region | Public | Community of Castilla-La Mancha | 46.716 M€ | 601–700 | 93 |
| Chinese Universities | Character | Autonomous Community/Province | GDP annual 2022 * | Shanghai Ranking ARWU | Number of Collaboration Agreements with Spanish Universities |
| Jinan, Shandong Province | Public | Shandong Province | 1.135 B€ | 901–1000 | 4 |
| Xuzhou, Jiangsu Province | Public | Jiangsu Province | 1.597 B€ | 1001+ | 5 |
| Shenzhen, Guangdong Province | Public | Guangdong Province | 1.678 B€ | 151–200 | 6 |
| Shanghai | Private | Shanghai | 0.580 B€ | 1001+ | 12 |

Source: GDP annual of Spain: https://datosmacro.expansion.com/pib/espana-comunidades-autonomas (accessed on 25 June 2024) GDP annual of China: https://www.sohu.com/a/655571605_100078606 (accessed on 25 June 2024). * Based on the example exchange rate of 1 CNY = 0.13 Euro.

When we contacted the participants, we sent them an interview outline (see Section 3), also enclosing information related to the research summary, confidentiality pledge, and the informed consent form; all participants were informed and confirmed consent by writing. Some interview questions referred to the methodology of the QS World University Rankings Best Student Cities Ranking [89]. These questions aimed to gain insights into the establishment of connections with international educational providers or institutions, serving six main purposes: (1) compensating for shortages of resources, funding, and expertise; (2) sharing knowledge, information, and technology; (3) increasing opportunities for the international mobility of students and staff members; (4) expanding the potential for course and curriculum development; (5) conducting collaborative research; and (6) providing joint programs and benchmarking [90]. Interviews and questionnaires were delivered in the native language of each participant. We sent each participant the questionnaire and interview invitation by email or WeChat and conducted the final interview through Tencent or Zoom meetings. The interview outline included the questions in Table 3. Although we have stated in the table that the different questions address different research objectives, all questions are relevant to the objectives under study (See Appendix A for questions).

**Table 3.** Questionnaire parts and items.

| Parts | Items | Total |
|---|---|---|
| Identifications | 1–7 | 7 |
| Covering our FIRST research objective: To understand the factors influencing Chinese students who choose to study in Spain, the obstacles to development, and the need for collaboration between Chinese and Spanish universities for Chinese students. | | |
| The factors influencing Chinese students coming to Spain to study. On a scale of 0–10, how would you rate the impact of the following factors on Chinese students studying in Spain? What are the scores? What are the reasons? | 8–14 | 7 |
| Obstacles to cooperation between Spanish and Chinese universities | 15–22 | 8 |

**Table 3.** *Cont.*

| Parts | Items | Total |
|---|---|---|
| What types of collaborations do Chinese students and universities need with Spanish universities? | 23–27 | 5 |
| Covering our SECOND research objective: To learn about the internationalization of higher education as it affects local development in China and Spain. | | |
| How can cross-border education contribute to regional development? How would you rate the following points on a scale of 0–10? What are the scores? What are the reasons? | 28–31 | 4 |
| Covering our THIRD research objective: To seek experts' views on cross-border distance education and future trends in the internationalization of education in China and Spain using new technologies | | |
| How do you see Chinese students and universities working with Spanish universities? | 32–35 | 4 |
| Total | | 35 |

Source: Authors.

## 4. Results

The answers provided by the eight interviewees to the questions are detailed below.

### 4.1. The Factors Influencing Chinese Students to Study in Spain

We followed the evaluation factors of the QS World University Best Student Cities Ranking methodology [89] and discussed with our experts whether these factors have a considerable impact on students' choice to study and live in Spain. According to the results, most factors are relevant, such as the university's prestige. However, issues related to culture and language are not so important, although most Chinese students choose to return to China after graduation, and they assign more value to the help provided by Spanish universities to find a job in China than to find one in Spain.

In Table 4, we observe that the experts' responses, with an average score of 7.87, reveal that university rankings and reputation play an important role in international cooperation and attracting students, particularly from countries such as China. Some experts emphasized the importance of university rankings in demonstrating the reputation of universities and promoting cooperation. In contrast, others stressed their broader implications for the future competitiveness of a country and the challenges posed by the current state of the university system. Overall, they agreed that improving rankings and reputations could increase opportunities for collaboration, making universities more attractive to prospective students. However, challenges, such as the limited choice available to students and the need to improve the university system in certain countries, such as Spain, still remain. This stresses the importance of addressing these challenges while using rankings and prestige to promote international cooperation and improve the quality of higher education.

Table 5 reveals that some experts considered the impact of studying with Chinese peers neutral (5 points) or low (Negative, 4 points) due to social integration challenges and students' varying preferences. In contrast, others emphasized the benefits of an international environment. This suggests that different experts and organizations have differing views on the impact of studying with Chinese peers, with some experts highlighting the importance of diversity and intercultural learning.

**Table 4.** Experts' opinions on the relevance of world rankings reflecting the prestige of the university.

| Expert | Score | Reasons |
|---|---|---|
| ES A | "I would probably score it as 7 or 8." | "We are private universities; from my point of view, Chinese students do not identify with private universities because of cultural issues; they trust public universities more. Therefore, a good world ranking is very important for us to show the prestige of our university, just like the University in the Catalonia and Madrid region World Marketing Ranking, which Chinese students easily favor." |
| ES B | "Yes, 8" | "University in the Valencian Region: We are the first technological university in Spain in ARWU Ranking 2023; however, the Spanish system is seriously regressing, while the German, French and Italian systems are advancing. Spain has spent many years without facing the real challenges of the university system and cutting funding in favor of private universities. Without a strong university system, Spain has no future. It is time to act with ambition, and one option is to collaborate with China. We cannot wait any longer." |
| ES C | "Absolutely important, 9" | "University in the Aragon Region was ranked 432 in the 2019 QS World Ranking. According to this ranking, it is easy to find collaboration opportunities with China." |
| ES D | "Absolutely important, 9" | "China has raised the profile of its universities through internationalization, so institutional representatives want to work with better universities, including students' interests." |
| CN E | "I would probably score it as 7 or 8." | "Although we collaborate with the best Spanish universities, students must choose a university according to their abilities." |
| CN F | "Absolutely important, 10" | "Prestige determines the students' attractiveness of a university, in addition to academic ability." |
| CN G | "Is 6" | "Students can only choose between the partner universities, so our students have few options. However, they prefer to go to the best-ranked one if they have several options." |
| CN H | "Yes, 7" | "Our biggest obstacle in working with Spanish universities is that there are very few QS Ranking top 500 and no QS Ranking top 100 universities." |

**Table 5.** Experts' opinions on the relevance of the students' nationality on the INHE.

| Expert | Score | Reasons |
|---|---|---|
| ES A | "I think it could be 6." | "It depends on the situation and whether a good student who speaks the language well wants to be in a class with a foreign student. However, in recent years, many students want to participate in a closed group of Chinese students. It is requested that the university offer them more help, such as with Chinese language assistance." |
| ES B | "I would probably score it as 6." | "In the academic year 2021/2022–2022/2023, the Master's in Business Innovation course received more or less 100 Chinese students; they do not care if the students studied with those of other nationalities; but, if possible, we want students from different nationalities to learn together, in an intercultural manner as part of the future of education." |
| ES C | "About 5" | "Chinese students are still not socially integrated; they work alone with other Chinese peers. So, for a student who knows the rest of the world, it doesn't have a particularly big impact on them." |
| ES D | "About 5" | "Chinese students don't integrate well socially; they all prefer to be with people from their own country." |
| CN E | "About 5" | "In the case of our graduates studying for a master's degree in Spain, most classes were 70% Chinese." |
| CN F | "About 5" | "We don't care." |

**Table 5.** *Cont.*

| Expert | Score | Reasons |
|--------|-------|---------|
| CN G | "Negative, 4" | "Students think that as long as they study at a foreign university, their peers will be foreigners; this is not a point to be considered." |
| CN H | "Yes, 9" | "Our students are more international and prefer to study in an international environment." |

In Table 6, the responses reveal that while some experts consider Spain's attractiveness less important than other factors, such as the city's reputation or the university's ranking, others consider it to be crucial. The attractiveness of Spain as a place to live, including factors such as safety and quality of life, varies in importance among the experts; according to them, the average score was 7.63, but it was generally considered to be a critical factor in attracting international students and promoting international cooperation. Even so, one expert in Spain gave only a pessimistic score of 4, arguing that Chinese students are concerned about the world ranking of their university more than anything else, even if they go to an unsafe place.

**Table 6.** Experts' opinions on the relevance of Spain's livability (i.e., the Economist Intelligence Unit's Global Livability Index, Safety, Health, and Well-Being).

| Expert | Score | Reasons |
|--------|-------|---------|
| ES A | "I think it's a 4. Chinese students don't care." | "For students who have not yet arrived in Spain, it is not the most important thing to choose a university or place of study: the first choice is always the world university ranking." |
| ES B | "I would probably score it as 7." | "Valencia, one of the most beautiful cities on the Mediterranean coast of Spain, is now the capital of Erasmus. The University of Valencia and the Politècnica University of Valencia might receive more than 5000 students from other countries in 2022–2023." |
| ES C | "I would probably score it as 7." | "Barcelona attracts many Chinese students because the city represents the brand of Spain." |
| ES D | "Absolutely important, 9" | "The University in the Castilla de La Mancha Region, its campus is located in a city that is mostly a World Heritage Site; the city is safe, and it is more convenient for parents to have their students come here." |
| CN E | "Yes, 7" | "Students prefer coastal cities such as Barcelona and Valencia." |
| CN F | "Absolutely 10" | "Spain obtains a perfect score in all habitability indices." |
| CN G | "Yes, 7" | "Girls, in particular, are more concerned about safety." |
| CN H | "Absolutely 10" | "Spain's habitability is the best help to make international cooperation possible." |

Table 7 shows that while most experts agree (a median score of 7.00) that job prospects in Spain are important, perceptions of their importance relative to other factors (such as university reputation, career goals, and familiarity with Spain) vary; in particular, one expert from Spain gave only a 4 because the target country for employment of Chinese students is still China. Another Chinese expert gave a 6 because the target position for employment of Chinese students in China is civil servants, for which the relevance of the degree to the position is more important. However, efforts to increase employment opportunities through internship programs were considered valuable in attracting and retaining international students.

**Table 7.** Experts' opinion on the relevance of future employment.

| Expert | Score | Reasons |
|---|---|---|
| ES A | "I think it's a 4." | "Our students, more or less 80% of Chinese students, will return to work in China; they only care about world rankings because the university's ranking will be an important factor for them to settle in a good city in China." |
| ES B | "I would probably score it as 7." | "Because in our lifelong learning center at University in the Valencian Region, the most sought-after master's degrees are those that can guarantee or benefit employment." |
| ES C | "This is important; I score it as 7." | "We had hoped to offer Chinese students a master's program + paid internship through University in the Aragon Region in collaboration with logistics companies in the Autonomous Community of Aragon, a very attractive project." |
| ES D | "I would probably score it as 6." | "This depends on the sector. I know that in the agricultural and environmental sector in Castilla La Mancha, there is a lot of work; maybe in the industrial sector, there are some places that don't have much work, but you can find other universities, for example, Barcelona, which has industrial estates offering more job opportunities." |
| CN E | "I would probably score it as 5–6." | "Students still want to go back to China in the future and apply for civil service examinations there; for them, a good degree is more important." |
| CN F | "About 7" | "The Chinese know less about Spain unless they study Spanish." |
| CN G | "I would probably score it as 7." | "Students can decide whether to stay and work abroad during their studies abroad." |
| CN H | "Absolutely important, 9" | "In Spain, finding work and obtaining a work permit is relatively easy." |

Table 8 reveals that affordability is important in attracting Chinese students to study in Spain. Although there may be differences in perceptions of the exact extent of the impact, the experts generally agreed (the average score was 8.00) that Spain's competitive tuition fees and low cost of living make it an attractive destination for international, including Chinese, students.

**Table 8.** Experts' opinion on the relevance of affordability (i.e., tuition, living expenses).

| Expert | Score | Reasons |
|---|---|---|
| ES A | "Yes, it has some impact: 6" | "For Chinese students who can study abroad, their family already has sufficient financial means." |
| ES B | "Yes, 6" | "Although tuition fees for official degrees are cheap because the government subsidizes them, more people obtain our degrees even though they are more expensive than official degrees because they offer good value for money." |
| ES C | "About 5 points, at the most 7 points" | "Zaragoza is a city that lives well; it doesn't have such a high cost of living as Barcelona; the tuition fee is an average level in Spain for foreigners, but when we talk to Chinese students who want to live and study in Madrid or Barcelona, they find it expensive. The general level of consumption in Spain is more or less the same, and Madrid and Barcelona, although more expensive than other cities, are similar and affordable for everyone. Of course, some students will still look for cheaper places." |
| ES D | "Yes, 9" | "For Chinese and other foreign students, the tuition fees are the same as for Spanish students: around 800–1000 euros/year. The cost of living in La Mancha is very low, so many international students choose to come to University in the Castilla de La Mancha region." |

**Table 8.** *Cont.*

| Expert | Score | Reasons |
|---|---|---|
| CN E | "Absolutely 9" | "It is Spain's competitive advantage." |
| CN F | "Yes, 9" | "Very good value for money is essential." |
| CN G | "Yes, 9" | "Affordability is the first factor that affects students." |
| CN H | "Absolutely 10" | "Spain's tuition fees and cost of living are relatively low within Europe, which makes it a very good option for our students." |

Table 9, with a median score of 7.00, reveals that while opinions vary on the importance of learning Spanish for Chinese students studying in Spain, its potential benefits, including cultural understanding, career opportunities, and ease of integration, are recognized. Additionally, one expert scored 4, and another Spanish expert (5 points) pointed out the importance of English language teaching for the INHE. Some experts highlighted the challenges of learning Spanish, particularly its grammatical complexity, while others emphasized the cultural and linguistic similarities between Spanish and Chinese that can facilitate acceptance and learning.

**Table 9.** Experts' opinions on the relevance of culture and language (cultural identity and learning Spanish—a new language).

| Expert | Score | Reasons |
|---|---|---|
| ES A | "Not a particular factor, 5" | "There are implications for students whose degree is taught in Spanish, but an increasing number of students are choosing majors taught in English." |
| ES B | "Yes, 7" | "Yes, at university, they still don't have many official studies taught in English or Chinese; learning a new language can be quite a challenge." |
| ES C | "Not a lot, 4" | "They don't care if you study a new language; they mostly search for a program taught in English, except students pursuing degrees in Hispanic philology or similar." |
| ES D | "Yes, 7" | "Here, language and cultural identity are very important. Spanish culture attracts people in China who desire to visit Spain. However, if they do not have enough proficiency in Spanish, this can pose a challenge." |
| CN E | "Depending on the situation 6" | "It should be based on the future direction of work and the life people want to choose." |
| CN F | "Yes, 7" | "Spanish is more difficult from a grammatical point of view." |
| CN G | "Yes, 8" | "Spanish is very important for the careers of our students." |
| CN H | "Absolutely 10" | "Spanish pronunciation is very similar to Chinese pinyin, and some cultures are very similar, so Chinese students easily accept it." |

In Table 10, the average score is 7.12, revealing that while scholarships are generally perceived to be important in attracting Chinese students to study in Spain, there are challenges such as eligibility restrictions, policy changes, and limited availability that may affect their effectiveness. Government support and favorable immigration policies are crucial in facilitating Chinese students' enrolment and enhancing their learning and living experience in Spain.

**Table 10.** Experts' opinions on the relevance of the role of the government (foreign policy).

| Expert | Score | Reasons |
|---|---|---|
| ES A | "No, 4" | "Chinese students cannot apply for scholarships and enjoy more political advantages, but they do not value them." |
| ES B | "Yes, 7" | "Chinese Service Center for Scholarly Exchange is not going to accredit Master's in Lifelong Learning. We have lost many students in self-study." |
| ES C | "About 8" | "Support at the government level is very important. In 2022, a large proportion of Chinese students who updated their Identity Card for Foreigners Student study permit with their undergraduate or master's degree transcript after studying Spanish at a Cervantes-accredited language center in Madrid have been rejected and have had to spend more time resolving their status issues, which also affects their learning and living experience." |
| ES D | "Yes, 7" | "I think it's easy to do the paperwork; you know better than I do that the paperwork is easy, yes, and if they go to other countries maybe it will be more complicated. So, the paperwork is simple in Spain." |
| CN E | "Yes, 9" | "The decrease in the number of Chinese students coming to Madrid to pursue university degrees in recent years is due to the admission policy." |
| CN F | "About 7" | "In China, low priority is given to the different types of scholarships to study in Spain." |
| CN G | "About 6" | "Generally, there are no scholarships." |
| CN H | "About 9" | "Chinese scholarship for students and the Spanish immigration policy is very favorable for the Internationalization of higher education institutions between China and Spain." |

*4.2. Obstacles to Collaborations between Chinese and Spanish Universities for Chinese Students*

When asked how they viewed obstacles to the INHE, five experts were optimistic, and three were pessimistic because they believed that cooperation between China and Spain is currently only at a preliminary stage.

Expert ES B said, "Our university is very happy to collaborate with China, but currently, we don't have official courses taught in English; thus, even if Spain is not a preferred destination country, they are going to freak out. Regarding the non-approval of the Master in Lifelong Learning (MFP), it is a pity because MFP had more Chinese students in the last two years than official courses. We admit Chinese students to official courses, such as official master's courses; we do require foreign students to have a Spanish B2 level accreditation. It is currently not necessary at the official degree level, but Chinese students have difficulties overcoming the cut-off mark for access to a degree".

Another pessimist, Expert ES C, stated, "Another obstacle of support from university management currently pertains to the collaboration model between Chinese and Spanish universities, which is still very basic compared to that of the United Kingdom and the United States. For example, educational exports to establish campuses in other countries and implement double degree programs in China almost do not exist. We understand that those who need the support of the university management to promote this type of official program, together with the National Agency for Quality Assessment and Accreditation (ANECA), since our university can no longer collaborate with China, they will have even more difficulty".

Expert ES D, who is also slightly pessimistic, said, "Difficulties related to academic collaboration between Chinese and Spanish universities include obstacles such communication and bureaucracy problems, lack of flexibility of public universities, shortage of programs taught in English, and the lack of fluent information".

Experts CN E, CN F, CN G, and CN H were optimistic. Although some obstacles still exist, such as the difficulty of mutual credit recognition, lack of social integration of Chinese students, and lack of enthusiasm of Spanish universities for cooperation, Chinese students and universities are confident in their cooperation with Spain, a central factor in students'

choice to study abroad [91]. Many restrictive policies have their rationale, such as China's non-recognition of cross-border distance learning and Spanish language requirements to ensure the quality of international education. Expert ES A, drawing on the successes of collaborations with exceptional Chinese students, highlighted that while the issues of social integration and language proficiency persist, they are not obstacles. However, the problem of not having an adequate number of courses taught in English can be resolved over time.

Thus, government policies and university support are indeed the main obstacles; as for other factors, most experts believe that admission policies, language, and cultural issues do not pose major challenges, although they do influence students' choices.

### 4.3. Types of Collaborations Chinese Students and Universities Need with Spanish Universities

All eight interviewees agreed that it was essential to determine the needs of Chinese universities and students. Expert ES C said, "Many Chinese universities are looking to collaborate with us, but in the end, the project does not have good acceptance because there are not enough Chinese students involved. From our experience with China, many students seek programs taught in English, but some feel that those taught in Chinese are better".

According to Chinese and Spanish interviewees, post-COVID-19 trends indicate that Chinese students prefer participating in programs with foreign teachers directly in Chinese universities, especially in an international campus model. Chinese–foreign cooperation in running schools and double degrees for bachelor's and master's courses are needed in China. Expert ES D contends, "Yes, we can do it, but we need better planning and more time, as the government approval process is very long both in China and Spain".

According to Expert ES C, an official master's degree taught in Chinese may become very popular. For example, the master's degree at CEDEU University Centre, which is attached to Rey Juan Carlos University of Madrid, taught in Chinese and approved by ANECA, had 80 Chinese students enrolling every year. However, it should not be taught in Chinese because studying abroad is a good opportunity to become proficient in a different language. Many Chinese universities plan to introduce a closed group of official master's degrees taught in English (Expert ES A). According to all Chinese interviewees, almost all professors from Chinese universities want to collaborate on doctoral studies with Spanish universities. Expert ES D said, "University in the Castilla de La Mancha region is helping Chinese professors pursue doctoral studies". During COVID-19, many Chinese agencies negotiated with Centro Formación Permanente de University in the València Region to create a headquarters or campus in China, according to Expert ES B. Expert CN F noticed a "new trend post COVID-19, not only of student mobility but also increased research collaboration". Expert CN E pointed out that the field of Education for Innovation and Entrepreneurship is the most interesting for Chinese students.

The Chinese university where CN Expert H works offers a three-year degree/diploma, attracting students to conduct projects to switch from a university to a four-year degree in Spain.

### 4.4. INHE Contributes to Local Development

We build on the importance of local development and learning skills mentioned in the literature review. All experts agree that the INHE can have positive consequences regarding the distribution of educational resources and regional economic development, with the intermediary role being especially important. Although most experts agree that technology is useful for INHEs, there is still some controversy about using innovative technology, with differing opinions.

Table 11 reveals that the INHE is generally perceived as valuable (the average score is 7.63) in facilitating access to quality education, sharing resources, and promoting cooperation between countries. However, there are differing views on the potential challenges and impacts, particularly regarding resource allocation and maintaining quality education within countries. For example, one Spanish expert was more neutral (5 points), and the

fact that Spain has helped to allocate resources to education in China also means that Spain needs to invest more resources in the arrival of Chinese students. Nevertheless, most experts recognize the potential benefits of the INHE in addressing educational inequalities and providing students with opportunities to access different academic disciplines and educational resources worldwide. In particular, ES B believed that the international campus model of cooperation not only solves the problem of distribution of quality resources but also promotes local development.

**Table 11.** Experts' opinions on alleviating the unequal distribution of educational resources.

| Expert | Score | Reasons |
|--------|-------|---------|
| ES A | "I would probably score it as 6–7." | "In relative terms, it can alleviate the problem of unequal distribution of educational resources in China, where it is difficult for people to access good Chinese universities. It could allow Chinese students to see the outside world and study in a foreign university." |
| ES B | "Yes, 7" | "Yes, we have more free space and more courses left. We can share this resource with China. Just as many foreign universities have plans to set up international campuses in China, constructing a new university can lead to developing various industries in the neighborhood, such as real estate, catering, employment, and so on." |
| ES C | "About 5" | "Yes, it is a way that can alleviate the unequal distribution of educational resources, but it may require adjustments. If more Chinese students come, the Spanish government should increase investment in the education sector to ensure that the public education system can provide high-quality education, benefiting Spanish citizens." |
| ES D | "Yes, 8" | "Yes, it can facilitate the distribution of quality educational resources." |
| CN E | "Yes, 8" | "Yes, in different countries, there are advantages in various academic disciplines. Through the INHE, students can choose the best specializations in different countries." |
| CN F | "Absolutely 10" | "The entrance exams for undergraduates or postgraduates are too competitive. The INHE provides an opportunity to study." |
| CN G | "Yes, 7" | "As shared resources." |
| CN H | "Absolutely 9" | "Quality education is mostly concentrated in economically developed areas and, through the INHE, more students have access to better quality education." |

Table 12, with a median score of 7, reveals that digital educational technologies have the potential to positively impact international cooperation between Spain and China by enhancing learning experiences, improving collaboration, and reducing costs. Although one expert gave a score of 4, there are differing views on the maturity of these technologies and the extent to which they should be relied upon, with some experts expressing caution and others highlighting their benefits; as experts ES B and ES D explained, Spanish universities are improving their institutions' services systems, and the use of online administrative systems and online resources systems will also be a very important step to accelerate the internationalization of education. Especially regarding the online service system, a paperless office will be of great help towards environmental protection. Certainly, according to the Chinese expert CN G (neutral (5 points)), technology has brought a lot of convenience, but the issue of the cost of the application of technology also still needs to be considered.

**Table 12.** Experts' opinions on harnessing technology for sustainable internationalization, innovation, and technology.

| Expert | Score | Reasons |
|---|---|---|
| ES A | "I would probably score it as 4–5." | "Negative. There are still some limitations for China, which should continue to allow the use of virtual platforms from other countries." |
| ES B | "Yes, 7" | "During COVID-19, we conducted live courses with ICT systems, which are the same quality as face-to-face classroom courses. Digital is to help students to solve the language barrier and make it more convenient. For example, our university's online student system, in addition to having a similar catechism system and the digital secretariat, students can use the digital secretariat to manage all the academic information, apply for transcripts and graduation certificates, with just one click can be applied at any time in any place, which is very convenient for foreign students, no longer need to be offline or through the mail to solve their administrative problems." |
| ES C | "About 8" | "Through metaverse, virtual reality, etc., maybe students learn better than in the classroom because those options are more fun, saving travel time and reducing the cost of international cooperation." |
| ES D | "Yes, 9" | "I think that new technologies have made it easier for two people to have online classes despite the different timetables. This reflects an improvement and serves as a tool that encourages more collaboration. To be able to help students studying online in other countries to have access to the school's resources, we have developed a complete system of online school resources, which are accessed through the VPN system of the student's account, such as the Student Administration System, databases, online libraries, and academic journal systems." |
| CN E | "I would probably score it as 6." | "The technology is not yet mature enough and needs to develop further. For example, for the popularization of the Digital Credentials system, such a system can guarantee that our students can obtain authentic certificates, because in international cooperation, due to the problem of asymmetric information, many students are worried that the signed documents they obtain are fake, but if Digital Credentials similar technology is further popularized, it will be a further guarantee for internationalization." |
| CN F | "Yes, 7" | "China is still at a basic stage of awareness with regard to innovative technologies and digital education in higher education. In our international collaborations, we are currently using more classroom formats such as through Microsoft Teams and cloud software to send and save files." |
| CN G | "5" | "Despite technological advances, we want students to be able to live and learn in Spain, which will make a difference. In addition, we have tried to promote the online study model. Although technology allows students to study and complete their studies, because of the cost of technology, there is not yet a particularly big difference in tuition fees between online and face-to-face classes, so more students want to spend the same amount of money to study abroad." |
| CN H | "Yes, 7" | "Some technology, such as language technology, is now available to help students fully understand what they learn in a course when they do not have strong language skills. But over-reliance on technology should also be avoided." |

Table 13 reveals that multilingual skills, particularly Spanish, can provide Chinese students with significant advantages in the labor market and in the field of international

relations. While there may be different perspectives regarding the extent of this advantage and the specific contexts in which it applies, based on an average score of 7.13, there is a general recognition among experts of the importance of language skills and specialized skills in building competitive advantage and succeeding in the global marketplace. However, one Spanish expert was more neutral (5 points), and whether Spanish is in high demand in the Chinese job market still needs further exploration.

**Table 13.** Experts' opinions on developing multilingual and multidisciplinary human resources for national and regional needs.

| Expert | Score | Reasons |
|--------|-------|---------|
| ES A | "Maybe, 5" | "In recent years, China has faced a shortage of Spanish language talent advantage. I live in Shanghai and certainly understand why people would want to go to Europe or the United States where English is spoken, but, the current demand for Spanish language skills among Chinese professionals largely comes from South America." |
| ES B | "Yes, 7" | "I think that an intercultural person who speaks another language that is not commonly spoken in their country and has accurate knowledge will have a good future." |
| ES C | "About 8" | "Yes, I think a Chinese student with special skills and a very good command of Spanish will find it much easier to find a job (e.g., in Chinese state-owned companies in Latam that have several needs)." |
| ES D | "About 8" | "That is to say, the greater the proficiency in the teaching and learning skills and languages, the better one will be able to relate to the rest of the world." |
| CN E | "Yes, 7" | "People with one or two specific-differentiated skills will be significant impact on the future job market." |
| CN F | "About 7" | "Chinese students have limited multilingual skills compared to Europeans." |
| CN G | "Yes, 8" | "Multi-skilled talent development is a strategy." |
| CN H | "Yes, 7" | "Build your competitive advantage." |

For Table 14, the average score is 8, indicating that the international education industry plays a crucial role in the economic development and internationalization efforts of many countries. Experts highlight its contribution to GDP, employment generation, income generation, collaboration, and international cooperation. While there may be slight differences in perspectives on the extent of its importance, most experts recognize the significant impact of the INHE sector on global education and the economy.

**Table 14.** Experts' opinion on the Study Abroad Services as an industry sector contributing to local development.

| Expert | Score | Reasons |
|--------|-------|---------|
| ES A | "About 7–8" | "Yes, the international education industry is the economic pillar of many countries." |
| ES B | "Yes, 7" | "It is a sector that includes many elements such as language school, student recruitment agency, training course, real estate agency service, and visa service. Related activities can create employment and bring more foreigners to Spain, increasing everyone's income!" |
| ES C | "About 8" | "Many countries are now publicly recognized as vehicles for revenue generation through increased international student numbers, tuition fees, and transnational educational operations." |
| ES D | "About 8" | "The more collaboration and internationalization we have with China and other Asian countries or their technologies, through import and export, the more we can contribute to the local development of communities." |

**Table 14.** *Cont.*

| Expert | Score | Reasons |
|---|---|---|
| CN E | "Absolutely 9" | "It is noted that the INHEI industry accounts for a large proportion of GDP in countries such as the United Kingdom, the United States, Canada, and Australia." |
| CN F | "Absolutely 10" | "This sector is the most important driver of INHE development." |
| CN G | "Yes, 7" | "Agencies, as for-profit organizations, are more motivated to promote the INHE; they serve as facilitators. For example, there are many intermediary organizations engaged in international education in China, and this industry is already a very important service industry in China as well." |
| CN H | "About 8" | "The other industries that have sprung up around the INHE speak for themselves of its importance." |

*4.5. Trends in Collaboration of Chinese Students and Universities with Spanish Universities*

On this issue, the eight interviewees were optimistic of the enormous future potential. Expert ES D said, "Because of the current projects of alliances between Asia and Europe, and Chinese universities working with European universities, several Spanish researchers are invited to go to China. I see that there is a lot of potential and opportunity for collaboration. One of the most crucial aspects is that logic is always followed; it is an economic logic of making a profit, but it would have to be more about the quality of this collaboration rather than the quantity. Thus, this collaboration should be of quality and followed up and evaluated if it bears fruit". However, Expert ES C said that China has not committed to opening up the market for distance education services in the WTO's Trade in Services Agreement, and there are currently no other Chinese laws and regulations regarding the recognition of cross-border distance education for the time being. Hence, the conditions for INHE through digitalization are not yet ripe. From the perspective of cooperation modes, Chinese experts believe it would be a great success if more Spanish universities could accept Chinese students for doctoral studies. With China's economic development, people have a greater desire to improve their academic qualifications. Another mode is the international campus, where Chinese universities establish branches in Spain, and Spanish universities establish overseas teaching centers in Chinese universities.

New technologies, platforms, and tools to promote greater INHE are good, considering that some countries do not allow their students to take an online foreign course because of the government's guarantee of the education market and the quality of education in the country. However, all eight interviewees were very positive about innovative technology that will encourage the INHE. The role of agencies should include facilitating collaboration, although they should prioritize quality over quantity and work based on logic rather than economic gain.

**5. Discussion and Conclusions**

Internationalization has become a strategically important agenda for higher education institutions [92]. As the COVID-19 pandemic recedes, our attention has been directed to geopolitics, healthcare, economics, education, and social culture [93,94]. This study worked with eight INHE experts in China and Spain, addressing key issues, including understanding the factors influencing Chinese students' choice to study in Spain, identifying the obstacles and needs for collaboration between Chinese and Spanish universities, assessing how the INHE affects local development in China and Spain, and exploring the future trends in the INHE in both countries using new technologies. Although some experts' perspectives differed, this study's results and overall perspective were similar. It highlights the importance of quality and the role of technology in future collaborations while also filling a gap in research on the INHE in China and Spain, in particular, a reference for the mode that can be employed to attract a greater number of Chinese students to study in Spanish universities.

Our findings reveal that the factors influencing Chinese students to study in Spain are world ranking, affordability, and the role of government. The latter two provide students with increased opportunities to study in Spain, while the world ranking is a key consideration for both Chinese students and universities. Because Chinese students are seeking high-quality education, the ranking helps them identify the quality, work opportunities, and registration issues in China. However, Chinese universities seek collaboration with Spanish universities that are prestigious. However, there are still obstacles to collaboration, such as communication and bureaucracy problems, the lack of flexibility of public universities, the shortage of programs taught in English, and the lack of clear information. The attitudes toward collaboration between China and Spain are open and positive. This study reveals that, following the recession of the COVID-19 pandemic, an increasing number of universities and students want projects such as double degrees, international campuses in China, and access to Spanish universities by switching their universities. A particularly important cooperation project that needs to be carried out is for Chinese students to study for a doctorate at Spanish universities.

The INHE will promote local development through the distribution of educational resources. Through the INHE, students can access better quality education, acquire labor skills, become multilingual and multidisciplinary individuals, and participate in transnational work, thus contributing to the sustainable development of the economy and the fulfillment of SDG 4. Agencies play an important role to facilitate collaboration and foster the INHE, but they should prioritize quality over quantity. The INHE is a vital issue for Spain, especially attracting more foreign students to pursue higher education in Spain, which has a huge impact on the economic growth of Spain [61].

Furthermore, this study reveals that the use of new technologies will drive future INHE collaborations, such as the digital secretariat, online administrative management system, online resource system, etc., similar to universities, which are the digital guarantee tools for internationalization, especially the digital certificate system. The further development of language technology applications will break the language barriers in internationalization, which is the biggest problem in the internationalization of China and Spain, that Chinese students need to learn Spanish to study in Spain. At the current stage, our technology application cost is still too high, and the tuition fees for online and face-to-face courses are basically the same. However, as the technology cost decreases exponentially, the cost of online courses will also drop significantly. This will also stimulate students to complete foreign courses online. Although cross-border distance education is not yet recognized in many countries due to quality concerns, the use of technology will undoubtedly favor the INHE in the future, especially by reducing communication, transport, and living costs and increasing the diversity of classroom formats. INHE digitalization can further contribute to the achievement of Agenda 2030 and sustainable development, by reducing environmental pollution through virtual mobility [26], by providing more opportunities for education to people around the world through online learning, and by helping students succeed through technology-based monitoring of their learning status. At the same time, it is necessary to avoid students' over-reliance on the use of technology, such as excessive reliance on translation tools [32], and to further consider the problem of the digital divide; the technological progress varies in different regions, and it takes time for the popularization of digitalization [40,42]. Promoting the INHE through the use of technology remains challenging from the point of view of the current level of technological development, policies, privacy, academic perspectives, and so on.

## 6. Limitations and Suggestions for Future Research

The INHE, which is a future trend for education, was the focus of this study. We also could not interview more INHE core experts as part of the data collection because the interviews were not official due to privacy issues, with some universities even refusing to participate. Therefore, only eight experts were interviewed, and some of their views differed, or these views were not representative; the findings were broadly similar. The

universities we selected in China and Spain are currently located in economically developed regions. Therefore, with government permission, future studies should facilitate research with more participating universities and INHE experts in China and Spain to promote local economic and educational development. In addition, this article only discusses the situation of Chinese students studying in Spain and discusses Spanish students studying in China. It is also possible to investigate patterns of learning Spanish, motivations for coming to Spain to study, etc., with Chinese students as the main target group. The world is facing new challenges and new environmental, social, and governance (ESG) risks after the pandemic; although this paper mentions some problems and solutions, it would be a great innovation to explore and analyze them from this perspective specifically. Finally, the study in this paper is limited to China and Spain, and a comparative analysis between multiple nationalities and different cultures may yield more diverse results.

**Author Contributions:** Conceptualization, Y.Q., A.G.-A. and R.I.-F.; Data curation, Y.Q., A.G.-A. and R.I.-F.; Formal analysis, Y.Q., A.G.-A. and R.I.-F.; Funding acquisition, Y.Q., A.G.-A. and R.I.-F.; Methodology, Y.Q., A.G.-A. and R.I.-F.; Project administration, Y.Q., A.G.-A. and R.I.-F.; Resources, Y.Q., A.G.-A. and R.I.-F.; Writing—original draft, Y.Q., A.G.-A. and R.I.-F.; Writing—review and editing, Y.Q., A.G.-A. and R.I.-F. All authors have read and agreed to the published version of the manuscript.

**Funding:** This research received no external funding.

**Institutional Review Board Statement:** Not Applicable.

**Informed Consent Statement:** Informed consent was obtained from all subjects involved in the study.

**Data Availability Statement:** The data presented in this study are available on request from the corresponding author due to privacy.

**Acknowledgments:** We are very grateful to the experts recommended by the international relations offices of the Spanish and Chinese universities mentioned in this article who made this study possible.

**Conflicts of Interest:** The authors declare no conflicts of interest.

## Appendix A. Questionnaire for Experts

(1) Expert's personal data

    (a) University:
    (b) Position: Vice-rectorate/Department/Other
    (c) Level of educational qualification:
    (d) Knowledge area:
    (e) Gender: Male/Female
    (f) Do you agree to have your views posted on the article? Yes/No
    (g) How many Spanish/Chinese universities is your university currently collaborating with?

(2) The factors influencing Chinese students coming to Spain to study. On a scale of 0–10, how would you rate the impact of the following factors on Chinese students studying in Spain? What are the scores? What are the reasons?

    (a) world ranking (Prestige of Spanish universities)
    (b) Student mix
    (c) Motivation for studying in Spain (e.g., Economist Intelligence Unit's Global Liveability Index, safety, health and well-being)
    (d) Employer activity
    (e) Affordability (e.g., tuition fees, cost of living)
    (f) Culture and language (e.g., cultural identity, learn a new language: Spanish)
    (g) The role of government (Foreigners' issues)

(3) Obstacles to cooperation between Spanish and Chinese universities

    (a) Difficulties regarding the mutual recognition of credits and qualifications

    (b) Lack of English-medium programs at Spanish universities

    (c) Motivation and enthusiasm for cooperation

    (d) Spain's low status as a preferred destination country for Chinese students

    (e) Social inclusion and academic success of Chinese students

    (f) The Scholarship Service Center of the Ministry of Education of China's refusal to certify the qualifications of Master Formacion Permanente in Spain

    (g) Restrictions on foreign students in admission policies (e.g., some Spanish autonomous communities require foreign students to have a B2 level of Spanish accredited by National University of Distance Education (UNED) for admission to undergraduate programs).

    (h) Others

(4) What types of collaborations do Chinese students and universities need with Spanish universities?

    (a) bachelor's degrees no official

    (b) Student exchange programs

    (c) Course taught in English. In Chinese.

    (d) Sino-foreign educational cooperation programs

    (e) Others

(5) How can cross-border education contribute to regional development? On a scale of 0–10, how would you rate the following points? What are the scores? What are the reasons?

    (a) Alleviating the unequal distribution of educational resources

    (b) Technology use for sustainable internationalization, innovation and technology driven internationalization

    (c) Cultivating multilingual and multidisciplinary human resources needed by the country or region

    (d) Study abroad services is an industry, and the growth of an industry can help the development of a region

(6) How do you perceive Chinese students and universities working with Spanish universities?

    (a) What do you think the future holds?

    (b) How will internationalization evolve in the future, given today's innovations and technological developments?

    (c) China has not committed to opening up the market for distance education services in the WTO's Trade in Services Agreement, and other domestic laws and regulations have yet to temporarily provide for the recognition of cross-border distance education. With regard to the development of technology, do you think that China will be able to recognize cross-border distance education in the future?

    (d) What is the role of the agency/intermediary role?

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
