# Peer review of "Internationalization of Higher Education in China with Spain: Challenges and Complexities"

_education, doi:10.3390/educsci14070799_

Round 1

Reviewer 1 Report

Comments and Suggestions for Authors

The article raises the topic: Internationalization of higher education in the context of universities in China and Spain. The study is well structured. The design of the empirical study – semi-structured interviews – is presented fully and clearly. The factors influencing the internationalization of higher education between China and Spain are identified in the tables. The results are detailed and the tasks correspond to the results.

But in addition to the fact that the study follows completely academic requirements, its relevance and scientific weight is questionable. Cooperation between universities in China and Spain does not occupy any significant place in the scientific agenda of higher education researchers. The challenges associated with this cooperation are typical of the challenges in the field of education between universities in Asia and Africa and non-English-speaking countries. To what extent does science need another confirmation of these challenges through research? I would assume that the scientific value of such studies is not the most significant, but once again I cannot help but note the high quality of the design of the experiment and the article: a clear research structure: definition of the problem, description through work with literature, development of the design of the experiment. The results are summarized in easy-to-understand forms - tables. The discussion provides additional questions for reflection.

            Although the scientific significance of the study remains debatable, I would recommend publishing it at least as an example of an excellent empirical study.

Author Response

Comments 1: The article raises the topic: Internationalization of higher education in the context of universities in China and Spain. The study is well structured. The design of the empirical study – semi-structured interviews – is presented fully and clearly. The factors influencing the internationalization of higher education between China and Spain are identified in the tables. The results are detailed and the tasks correspond to the results.

But in addition to the fact that the study follows completely academic requirements, its relevance and scientific weight is questionable. Cooperation between universities in China and Spain does not occupy any significant place in the scientific agenda of higher education researchers. The challenges associated with this cooperation are typical of the challenges in the field of education between universities in Asia and Africa and non-English-speaking countries. To what extent does science need another confirmation of these challenges through research? I would assume that the scientific value of such studies is not the most significant, but once again I cannot help but note the high quality of the design of the experiment and the article: a clear research structure: definition of the problem, description through work with literature, development of the design of the experiment. The results are summarized in easy-to-understand forms - tables. The discussion provides additional questions for reflection.

Although the scientific significance of the study remains debatable, I would recommend publishing it at least as an example of an excellent empirical study.

Response 1: 

Thank you for recognising our article. We believe that the greatest significance of this article is to fill the gap in the research on international cooperation between Spain and China in higher education. We expect that this article will give certain references to Spanish and Chinese universities in international cooperation and scientifically establish a sustainable partnership so that students can have access to educational resources more suitable for their own and promote the local economy.

Reviewer 2 Report

Comments and Suggestions for Authors

Besides this study findings that are not incredibly disruptive, and the overall clarity of the manuscript, I am worried about the iThenticate report: 13% plagiarism detected is connected to different sources that are not always adequately referenced or appear too similar. 

I believe authors should consider removing this issue before considering its submission to a journal.

Comments on the Quality of English Language

Language is fine, but the use of some acronyms is often inconsistent.

Author Response

Comments 1 Besides this study findings that are not incredibly disruptive, and the overall clarity of the manuscript, I am worried about the iThenticate report: 13% plagiarism detected is connected to different sources that are not always adequately referenced or appear too similar. 

I believe authors should consider removing this issue before considering its submission to a journal.

Response 1: 

Thank you for raising this issue; we have amended some of the quoted examples. However, much of the same content is mostly specific data, different proper nouns, titles of policies, and specific content of policies, e.g., SDGs in full and abbreviated, specific policy content, etc.

Reviewer 3 Report

Comments and Suggestions for Authors

Internationalization of university education is a tool for maintaining stability in the world, developing friendly relations between countries. Therefore, the topic of the article is very important, especially since it deals with countries with different histories and cultures. The article is well structured, the logic of the presentation of the material is clear. The introduction provides an extensive overview of the current understanding of the importance and challenges of internationalization of education. The research methodology, which is based on the semi-interview method, is described in sufficient detail. However, the study has one significant flaw that calls into question some of the results and conclusions: very few experts were involved in the study. Therefore, the main comment relates to the number of experts.

1.       The authors do not provide arguments to assess the necessity and sufficiency of the opinions of eight experts to reveal the research topic. Moreover, in a number of tables, expert opinions differ significantly (for example, in Tables 5, 6, 7, 9, 12). Why can the conclusions drawn from the opinions of eight experts be considered representative? Is it possible that if the circle of experts expands, the data obtained will be different?

This remark justifies mistrust in the reliability of the article’s conclusions. Perhaps the authors should still work on finding other experts of the same qualifications and then return to the article. Here are a few more non-critical comments:

2.      It is noteworthy that in many places the article summarizes the conclusions of experts from the standpoint of a quantitative approach. The authors indicate that MOST experts hold one opinion or another. But the results of qualitative research are of little use for formulating quantitative conclusions. To make a quantitative conclusion, it is necessary to use a representative sample constructed on the basis of statistical calculations.

3.       In the Introduction it is stated that the authors say “we conduct an in-depth investigation into the INHE in both countries to find the most suitable cross-border higher education sustainability models after the COVID-19 era” (lines 84-86). However, such models are not presented in the article.

4.      The second objective  To explore  INHE and its impact on the local development of China and Spain” was not fully achieved, since the impact on the local development of China is practically not shown.

5.      In the section "INHE in the era of digitalization: positive or negative?" the authors focus on the advantages and disadvantages of digitalization of education in general, repeating the results repeatedly described in various publications. The title of the article leads us to expect conclusions from this section regarding the internationalization of education in China and Spain. But I have not noticed such conclusions.

Author Response

Comments 1:The authors do not provide arguments to assess the necessity and sufficiency of the opinions of eight experts to reveal the research topic. Moreover, in a number of tables, expert opinions differ significantly (for example, in Tables 5, 6, 7, 9, 12). Why can the conclusions drawn from the opinions of eight experts be considered representative? Is it possible that if the circle of experts expands, the data obtained will be different?

Response 1: Our final selection of respondents followed the recommendation of the international relations office because these experts are responsible for the Internationalization of Higher Education (INHE) between China and Spain; they have a very large number of successful and unsuccessful cases of cooperation between China and Spain. Of course, in the event that we could have had more respondents, we might have been able to get more perspectives; however, it is rather unfortunate that because of the scarcity of INHE between China and Spain, not a lot of people are aware of this area, so we have made this clear in our limitations.

Comments 2: It is noteworthy that in many places the article summarizes the conclusions of experts from the standpoint of a quantitative approach. The authors indicate that MOST experts hold one opinion or another. But the results of qualitative research are of little use for formulating quantitative conclusions. To make a quantitative conclusion, it is necessary to use a representative sample constructed on the basis of statistical calculations.

Response 2: Thank you for your question. We use quantification in the results section through average score, median, and special scores to provide evidence for the ideas, making the article more logical and especially more convincing.

Comments 3:In the Introduction it is stated that the authors say “we conduct an in-depth investigation into the INHE in both countries to find the most suitable cross-border higher education sustainability models after the COVID-19 era” (lines 84-86). However, such models are not presented in the article.

Response 3: We have added the names of the modes, such as, Chinese-medium programmes, international campuses, bachelor's degrees no official or Student exchange programmes, etc., while the international campuses mode has a specific mode description in lines 196-213.

Comments 4:The second objective “To explore INHE and its impact on the local development of China and Spain” was not fully achieved, since the impact on the local development of China is practically not shown.

Response 4: Thank you for your suggestions to improve our results, we have improved the expert's views in the Tables 11 and 14 sections of the results; for example, the construction of international campuses can lead to the development of other local industries, and intermediary organisations engaged in international education are also already one of the very important service industries.

Comments 5:In the section "INHE in the era of digitalization: positive or negative?" the authors focus on the advantages and disadvantages of digitalization of education in general, repeating the results repeatedly described in various publications. The title of the article leads us to expect conclusions from this section regarding the internationalization of education in China and Spain. But I have not noticed such conclusions.

Response 5: We have included the original statement of the Expert ES C " considering that some countries do not allow their students to take an online foreign course because of the government's guarantee of the education market and the quality of education in the country " in section 4.5 to make it clear that it is not yet realistic to realise the INHE between China and Spain through digitalization at this stage.

Reviewer 4 Report

Comments and Suggestions for Authors

The manuscript presents an interesting and up to date discussion. Since Mandarin and Spanish are the two most spoken languages it is interesting to validate the exchange programs between China and Spain. Nevertheless there are some items that should be clarify prior to publication:

1) the manuscript addresses the impact on Chinese speaker students going from China to university programs in Spain. It would be interesting to introduce at least some comments on possibly published remarks/references on Spanish speaker students going from Spain to university programs in China.

2) Mandarin and Spanish are very different languages and it requires interest and effort to learn it as a foreign language. Mandarin is already taught in Spain at secondary school level. Is it possible to know how Spanish is taught in China? Among those Chinese students that went to study in Spain what percentage studied Spanish prior to arrival? What was their previous interest on Spanish culture?

3) At the bottom of Tables 1, 2 and 3 it shows "Source: Authors.". Does that mean that the details can be asked to the authors or what?

4) "GDP per capita 2022" is one of the columns of Table 2. Are the numbers per capita? Where did the authors collected the numbers and what is the interest since no discussion is addressed along the text?

5) Experts' responses/opinions and "scores" are presented from Table 4 to Table 14. That makes 11 tables that correspond to 6 Parts and 35 items of the questionnaire presented at Appendix A.  Looking at the Questionnaire only Part (2) and (5) present items to be scored. Does that mean that tables 4 to 14 address only those 2 Parts?  Does that mean that the possible answers to the other (1), (3), (4) and (6) Parts are given along the text. Where? This situation is very confusing for the reader. Can the authors clarify what corresponds to what, Part by Part or Item by Item?

6) the manuscript is well written presenting very few typo errors like for instance on page  6 line 260 it shows Span and it should be Spain. Also the Appendix A presents words/sentences in Spanish. It is on purpose or forgot to translate to English? Please give a further review to the manuscript in case more typo errors are in.

Comments on the Quality of English Language

Very few typo errors were detected. For instance on page  6 line 260 it says Span and it should be Spain. Also the Appendix A presents words/sentences in Spanish. It is on purpose or forgot to translate to English?

Author Response

Comments 1: the manuscript addresses the impact on Chinese speaker students going from China to university programs in Spain. It would be interesting to introduce at least some comments on possibly published remarks/references on Spanish speaker students going from Spain to university programs in China.

Response 1: Thanks to the reviewer for this suggestion, which is a limitation of this article, we can dedicate a future research on Spanish students studying in China. I've added this suggestion to the limitations

Comments 2:Mandarin and Spanish are very different languages and it requires interest and effort to learn it as a foreign language. Mandarin is already taught in Spain at secondary school level. Is it possible to know how Spanish is taught in China? Among those Chinese students that went to study in Spain what percentage studied Spanish prior to arrival? What was their previous interest on Spanish culture?

Response 2: We've already researched the reasons why Chinese students choose to study in Spain based on the QS World University Rankings Best Student Cities Ranking methodology in section 4.1, but the question you posed is so interesting that we could write a new article could be written entirely to explain the description, so I've also made this clear in the limitations

Comments 3:At the bottom of Tables 1, 2 and 3 it shows "Source: Authors.". Does that mean that the details can be asked to the authors or what?

Response 3: Yes, as this form is our own designed and edited form, you can contact us with any questions.

Comments 4: "GDP per capita 2022" is one of the columns of Table 2. Are the numbers per capita? Where did the authors collected the numbers and what is the interest since no discussion is addressed along the text?

Response 4: We have added data sources to the table that are GDP annual, Also, In the article it is explained that in order to illustrate the economic variability of the regions where the universities under study are located, a comparison is made between regions with different economic levels, while we list the GDP annual of the regions where the universities are located.

Comments 5:Experts' responses/opinions and "scores" are presented from Table 4 to Table 14. That makes 11 tables that correspond to 6 Parts and 35 items of the questionnaire presented at Appendix A.  Looking at the Questionnaire only Part (2) and (5) present items to be scored. Does that mean that tables 4 to 14 address only those 2 Parts?  Does that mean that the possible answers to the other (1), (3), (4) and (6) Parts are given along the text. Where? This situation is very confusing for the reader. Can the authors clarify what corresponds to what, Part by Part or Item by Item?

Response 5: For Part (2) and (5) it is correct.Part (1) in Appendix A corresponds to Tables I and II, and because of the privacy issues involved, we only show what can be made public in the tables.

Part (3) corresponds to section 4.2, part (4) corresponds to section 4.3, and part (6) corresponds to section 4.5. The reason why we do not show the results based on the sub-questions one by one is that the experts answered these questions in a way that is often related to each other, and we summarised the final results by following the pattern of the experts' answers.

Comments 6: the manuscript is well written presenting very few typo errors like for instance on page 6 line 260 it shows Span and it should be Spain. Also the Appendix A presents words/sentences in Spanish. It is on purpose or forgot to translate to English? Please give a further review to the manuscript in case more typo errors are in.

Response 6: Thank you for your affirmation. Page 6, line 260 has been changed to "Spain". We've done an English translation and revised it on the article because when doing the interviews, some of the Spanish proper nouns were easier for the participants to understand.

Round 2

Reviewer 2 Report

Comments and Suggestions for Authors

Author Response

Comments 1: First, starting from the abstract content, the study lacks motivations regarding the decision to choose Spain and China for the analysis. It should be at least briefly introduced the need for exploring the present fields.

Next, the introductive section involves the same issues I described for the abstract. While authors clarify the effects of normative provisions for China and Spain, to me it is not clear why scholars should be interested in exploring the two countries’ activities. Specifically, I would like to read proper theoretical implications for exploring the field, together with more in-depth practical implications regarding sustainable change via Agenda 2030.

Response 1: Thanks to the reviewer for pointing this out; we have further clarified our research motivation in this revision in the abstract and introduction: we want to identify the differences and commonalities between the two countries by comparatively analyzing their higher education systems, quality assurance, internationalization strategies, and sustainability goals, to search for the most suitable post COVID-19 era cross-border higher education achievability models, such as East-West programs, international campuses, bachelor's degrees, no official or student exchange programs, etc., and to identify differences and commonalities between the two countries, as well as areas in which Spanish universities can improve, both to meet the needs of China and to contribute to the achievement of the Agenda 2030 SDGs, in order to allow Spanish universities to have articles of reference that will help the Spanish universities to attract more Chinese students.

Comments 2:Moving to the state-of-the-art section, I do understand that the study’s explorative nature requires a focus on grey literature, but authors should better connect it to scientific literature. Moreover, a table representation of the key themes for each paragraph could help enhance readability.

Response 2: You have given us a very good suggestion, we have compiled a table of INHE related reports, government policies, and legal documents that appear in the article in chronological order and placed it in Appendix B to help enhance readability.

Comments 3:Considering readability, I’d recommend authors to disclose whether AI tools were used to help improve language: I see that some sentences are too articulated or involve keywords that usually trigger AI detectors (e,g, the use of the verb “to delve”).

Response 3: We didn't use AI tools to help in the language, we just thought to delve might look more advanced than to explore.

Comments 4:Regarding objectives paragraph, I feel that the objectives are too wordy and a little deceptive. Specifically, I understand that the perspective is from China to Spain and not bidirectional: if so, authors should clarify it throughout the manuscript starting from the abstract.

Response 4: We have also made more specific motivations explicit in our research objectives.

Comments 5:Moving to methodology, authors state that one key issue is the number of participants eligible for the analysis. This issue must be solved by integrating the qualitative analysis using an enforcing data triangulation approach through the exploration of practice documents and reports available in the universities (e.g., sustainability reports). In this way, the analysis is not complete and offers only a partial exploration of the topic. In addition, the method limitations should also be addressed in the final section of the manuscript.

Response 5: We have made this clear in the limitations. Indeed, the findings would have been more meaningful if they had been analyzed further in a comparative manner from practice documentation and reports.

Comments 6:Subsequently, I go back to my comment presented in the first round of revisions. Specifically, I find that the study main issue is the lack of relevant implications. Reading the results and discussion, it emerges that factors as affordability, quality education and prestigious partners are key factors in influencing Chinese students: what is innovative about this? I feel that it is possible to change the nationality of the students and get the same results. Same to be said concerning obstacles (bureaucracy has always been an obstacle in any public institution), and I do not see how the debate contributes to the post-pandemic: the world is facing new issues and new ESG risks – that are not listed or explored – are emerging.

Response 6: Although this paper does not analyse from the Environmental, social, and governance (ESG) point of view, its content basically contains ESG risk issues, such as environmental risk, the use of technology to reduce environmental pollution brought about by transport, paperless office issues, social risk, but also from the advancement of economic development, promote employment, promote education for all, and to address the unequal distribution of resources in education, etc., from the governmental risk, geopolitical issues, and government policy issues.

Comments 7:Next: “Furthermore, this study reveals that the use of new technologies will drive future INHE collaborations”. Please elaborate it more, it looks like a standard statement to increase conclusions length. Also, implication for theory are very limited and not well presented, along with policy implications that should be central for this manuscript.

Response 7: Yes, this is the core of this article and its innovation, and we have refined the results and conclusions of this section separately, and also summarized them based on the content of the literature

Comments 8:Finally, while I do not see language issues, I recall my suggestion in previous sections: somehow the language appears heterogeneous, and it could be necessary to harmonise it on more homogeneous lexical choices.

Response 8: We have further optimized the language issue

Reviewer 3 Report

Comments and Suggestions for Authors

To the answer to the first comment.

As I wrote in the previous review, the article touches on a very interesting topic and contains all the signs a good scientific research. The problem is that the semi-interview method that authors chose for the study must meet the relevant methodological requirements. These requirements contain the conditions that research continues until a clear qualitative picture of material under study emerges. In some cases 8 experts are enough for this. In other cases this number is not enough. The qualifications of experts are not of decisive importance. The spread of estimates in some tables shows that 8 experts are not enough for these cases. Therefore, the authors ' conclusions for these cases cannot be considered reliable. The authors should do more work to attract experts or sacrifice these tables.

I can agree with the rest of the answers.

General conclusion. The article cannot be published in this form.

Author Response

Comments 1: As I wrote in the previous review, the article touches on a very interesting topic and contains all the signs a good scientific research. The problem is that the semi-interview method that authors chose for the study must meet the relevant methodological requirements. These requirements contain the conditions that research continues until a clear qualitative picture of material under study emerges. In some cases 8 experts are enough for this. In other cases this number is not enough. The qualifications of experts are not of decisive importance. The spread of estimates in some tables shows that 8 experts are not enough for these cases. Therefore, the authors ' conclusions for these cases cannot be considered reliable. The authors should do more work to attract experts or sacrifice these tables.

Response 1: Thank you for pointing out this problem, we also know that the number of interviewees is too small, as we said in the first round of replies, many Spanish universities do not have much experience in accepting Chinese students on a large scale, or many Chinese universities have not sent students to study in Spain on a large scale, and there are also privacy issues involved in the interviews that some school experts are not willing to accept. Even if we increase the number of interviewees to 10 or 12 and summarise the results, the general results may not change much, because in the answers given by these 8 experts, although they have many different points of view, they have the same objective, so the different points of view are not representative, which is also clear in the conclusions and limitations.

Round 3

Reviewer 2 Report

Comments and Suggestions for Authors

After revising authors work on my suggestion, I feel quite happy with the new version of the manuscript. I will leave it to the Editor for the decision on this manuscript.

Author Response

Comments 1: After revising authors work on my suggestion, I feel quite happy with the new version of the manuscript. I will leave it to the Editor for the decision on this manuscript.

Response 1: Thank you for your affirmation; your guidance, made this article able to be more valuable.

Reviewer 3 Report

Comments and Suggestions for Authors

Dear authors,

As I noted earlier, I really like the idea of ​​your article about research on internationalization in education and the research method you chose.

I am very sorry that you are unwilling to understand that the research method you have chosen can only provide a qualitative analysis of the problem. To get a good picture of each question you pose, you may need a different number of experts. In the tables I indicated, you received large differences in expert assessments. The average score that you use when analyzing tables when there are large differences in expert assessments shows absolutely nothing. This average score, as well as the words “most experts agree” when analyzing Table 7, cannot serve as evidence. Your comments on Table 5 look pretty good now. But similar  comments you can apply to tables 6, 7, 9, 11. Essentially, such comments indicate that it is impossible to draw any conclusions on the issues that are examined in these tables. Therefore, I see no point in publishing these tables. What scientific idea does such a publication convey?

That is why I am against publishing your article in the form in which it is currently presented.

If the editor finds my arguments insufficiently convincing, I will not object to his decision.

Author Response

Comments 1: I am very sorry that you are unwilling to understand that the research method you have chosen can only provide a qualitative analysis of the problem. To get a good picture of each question you pose, you may need a different number of experts. In the tables I indicated, you received large differences in expert assessments. The average score that you use when analyzing tables when there are large differences in expert assessments shows absolutely nothing. This average score, as well as the words “most experts agree” when analyzing Table 7, cannot serve as evidence. Your comments on Table 5 look pretty good now. But similar comments you can apply to tables 6, 7, 9, 11. Essentially, such comments indicate that it is impossible to draw any conclusions on the issues that are examined in these tables. Therefore, I see no point in publishing these tables. What scientific idea does such a publication convey?

Response 1: Thank you for your suggestion. We feel that it is very important and we have revised the comments on Tables  6, 7, 9, 11, 12 and 13 further to explain the negative and neutral views of the experts to make the results more critical. We did find more meaningful points again in the process of revising them. For  example, Table  7 provides a better understanding of the employment destinations and job positions of Chinese students. Table  9 offers the relevance  of English language teaching to promote the internationalization of higher education. Table 12 shows Cost issues of technology adoption, among other aspects.